# Mn-catalysed acceptorless dehydrogenative condensation of ureas with 1,2-diols for synthesizing imidazolones
Jiaqiao Ding[1], Di Kang[2], Tianshu Kou[2], Shan Lv[1], Siyi Song[1], Bing Cui [1], Mingqin Zhao[1] ✉ & Zhihui Shao [1] ✉

A plethora of biologically active compounds contain an imidazolones system as a central skeleton. Therefore, developing efficient methods for constructing such a skeleton holds significant research value. Here we show an efficient green procedure for synthesizing imidazolones via dehydrogenative condensation of urea with 1,2-diols. The reaction proceeded efficiently under mild conditions in the presence of a $^{Ph}$PNP-Mn catalyst and a weak base ($Na_2CO_3$). The applicability of the proposed catalytic reaction was highlighted by synthesizing more than 30 imidazolone derivatives, bearing different functional groups, in good to excellent isolated yields. Our study reports on the dehydrogenative condensation of ureas with 1,2-diols to synthesize imidazolones using a homogeneous non-noble metal catalyst. The proposed catalytic reaction proceeded even at a low catalyst loading of 0.05 mol%, with a high turnover number of 1660, resulting in yields up to 99%.

N-heterocycles are well known for their remarkable biological and pharmacological activities[1-9]. Among *N*-heterocycles, imidazolones and their derivatives hold significant value in the synthesis of natural products, such as dibromophakelstatin[10] and axinohydantoins[11]. They are integral components of several currently marketed drugs, largely because of their notable antiprotozoal and antiviral properties[12-18]. For example, loxoribine, a guanosine analog, is known for its antiviral and antitumor activities. It functions as a synthetic adjuvant for antitumor responses[19]. Another example is theacrine, a methylpurine derivative that strengthens the heart, promotes diuresis, dilates the coronary arteries, relaxes bronchial smooth muscle, and stimulates the central nervous system. It is primarily used to treat bronchial asthma, emphysema, bronchitis, and cardiac dyspnea[20,21]. Furthermore, imidazolone units form the core structures of important drug intermediates such as catramilast and metazamide[22] (Fig. 1).

Owing to the importance of imidazolones as a structural unit in various molecules and frameworks, many synthetic methods have been developed, including hydroamidation of propargylic urea[23-29], diamination of olefins or aryl ketones with diaziridinones[30,31], and diamine carbonylation[32-34]. However, these methods often suffer from low atom efficiencies, poor functional group tolerance, and inferior regioselectivities. Thus, research on developing new synthetic processes is highly needed for preparing highly functionalized imidazolones and their derivatives in an atom-economical and sustainable manner. Alcohol molecules are highly abundant, economical, and eco-friendly starting materials that can be produced from renewable bio-based feedstock[35,36]. In recent years, acceptorless dehydrogenation (AD) of alcohols catalyzed by transition metals has attracted great attention for its sustainable and atom-economical nature, producing only $H_2$ and $H_2O$ as byproducts, without needing stoichiometric reagents or $O_2$[37-48]. Alcohols have become crucial precursors for synthesizing *N*-heterocyclic chemicals via AD and condensation using transition metal catalysts[49-57].

Some reports suggest that diols are attractive, ecofriendly, readily available, and stable building blocks for synthesizing *N*-heterocyclics[52,58-63]. Synthesizing imidazolones using urea substrates and 1,2-diols via AD can be anticipated to be a versatile method. To the best of our knowledge, only a few reports on the synthesis of imidazolones using diols have been published. In 2022, Adam reported the first heterogeneous Pd-catalyzed protocol for the AD and condensation between *N*, *N*'-disubstituted ureas and 1,2-diols to afford imidazolones[64] (Fig. 2A). However, the previously reported catalytic systems have problems, such as difficulty in regulating the structure of active sites and extremely complex characterization. Notably, well-defined

[1]Flavors and Fragrance Engineering & Technology Research Center of Henan Province, College of Tobacco Science, Henan Agricultural University, Zhengzhou, China. [2]Technology Center of China Tobacco Hebei Industrial Co., Ltd, Shijiazhuang, China. ✉e-mail: zhaomingqin@126.com; shaozh21@henau.edu.cn

**Fig. 1 | Examples of natural products or drugs containing the imidazolones scaffold.** Loxoribine is known for its antiviral and antitumor activities, theacrine is a type of methylpurine drug, catramilast and metazamide are important drug intermediates.

Loxoribine    Theacrine    Catramilast    Metazamide

**Fig. 2 | Acceptorless dehydrogenative condensation of ureas with 1,2-diols. A** Heterogeneous Pd-catalyzed synthesis of imidazolones from ureas and 1,2-diols. **B** Homogeneous Ru-catalyzed synthesis of imidazolones and oxazolidin-2-ones from ureas and 1,2-diols. **C** Mn-catalyzed synthesis of imidazolones from ureas and 1,2-diols.

**A) Heterogeneous catalysis:**

Adam et al. *ACS Catal.* 2022, *12*, 6906-6922.

**B) Homogeneous catalysis:**

Watanabe *et al. J. Chem. Soc., Chem. Commun.* 1992, 1318-1319.

Beller et al. *Angew. Chem. Int. Ed.* 2016, *55*, 7826.

**C) This work:**

>25 examples, up to 99% yield

• **Base-metal catalysis**  • **Broad scope**  • **Highly selective**

homogeneous catalysts are potentially more active under high selectivity and milder conditions that can be tuned based on insights gained from mechanistic studies, such as work by Watanabe on homogenous RuCl$_2$(PPh$_3$)$_3$ catalyzed synthesis of imidazolones from N, N'-disubstituted ureas and 1,2-diols[65]. In 2016, Beller et al. reported a domino process for Ru-catalyzed synthesis of oxazolidin-2-ones from ureas and vicinal diols[66] (Fig. 2B). Note that only a handful of transition metal catalysts that can efficiently promote the synthesis of imidazolones from ureas and diols have been developed; however, most of the developed catalysts are restricted to noble metal catalysts, mainly based on ruthenium. For sustainability, replacement of noble metal catalysts with economical and environmentally benign earth-abundant metals is in high demand[67,68]. Therefore, atom-economic dehydrogenative condensation of urea with 1,2-diols to form imidazolones catalyzed by a non-noble metal catalyst is highly desirable. Since 2016, a series of novel Mn pincer catalysts have been successfully developed for use in both hydrogenation[69–71] and dehydrogenation[72–78] reactions. Encouraged by the significant achievements in the development of non-noble metal catalysts for synthesizing application[79–85], we developed the Mn-catalyzed alcohol-based AD and condensation of urea with 1,2-diols (Fig. 2C), affording a wide range of imidazolones (>30 examples), with high selectivity and efficiency (up to 99% yield and a turnover number (TON) of 1660).

## Results and discussion

### Optimization of the reaction conditions

Based on the development of Mn pincer catalysts[86–96], we investigated the catalytic activities of Mn PNP pincer complexes **I–IV** (Table 1). The reaction between N, N'-dicyclohexylurea **1a** (0.5 mmol) and 2,3-butanediol **2a** (1 mmol, 2 equiv.) using 1 mol% $^{iPr}$PNP complex [**Mn**]-**I** as a catalyst and 10 mol% $^t$BuOK in 0.5 mL dioxane as a solvent in a closed Schlenk tube at 160 °C for 16 h results in forming 1,3-dicyclohexyl-4,5-dimethyl-1,3-dihydro-2H-imidazol-2-one product **3a** in 14% yield, as revealed by GC-MS analysis (Table 1, entry 1). The use of $^{cy}$PNP complex [**Mn**]-**II** as a catalyst afforded **3a** in 9% yield (Table 1, entry 2). Changing the catalytic system to [**Mn**]-**IV** supported by bulkier $^{tBu}$PNP ligands afforded **3a** in a 6% yield (Table 1, entry 4). Notably, the use of a pincer complex supported by the $^{Ph}$PNP ligand [**Mn**]-**III** as a catalyst afforded **3a** in 62% yield (Table 1, entry 3). Note that the use of commercially available MnCl$_2$ or Mn(CO)$_5$Br as a catalyst yielded no products, indicating the importance of the supporting carbonyl and pincer ligands in the catalytic systems (Table 1, entries 6 and 7). The reaction did not proceed in the absence of Mn as a catalyst (Table 1, entry 5). Comparing the reaction outcomes, [**Mn**]-**III** can be considered to be an efficient catalyst for the dehydrogenative condensation of N, N'-dicyclohexylurea (**1a**) with 2,3-butanediol (**2a**).

**Table 1 | Optimization of reaction conditions**

| Entry | [Mn] | Base | Solvent | n [mol] | m [mL] | x [mol%] | T [°C] | Y$_{3a}$ [%] |
|---|---|---|---|---|---|---|---|---|
| 1 | [Mn]-I | $^t$BuOK | dioxane | 1 | 0.5 | 10 | 160 | 14 |
| 2 | [Mn]-II | $^t$BuOK | dioxane | 1 | 0.5 | 10 | 160 | 9 |
| 3 | [Mn]-III | $^t$BuOK | dioxane | 1 | 0.5 | 10 | 160 | 62 |
| 4 | [Mn]-IV | $^t$BuOK | dioxane | 1 | 0.5 | 10 | 160 | 6 |
| 5 | none | $^t$BuOK | dioxane | 1 | 0.5 | 10 | 160 | trace |
| 6 | MnCl$_2$ | $^t$BuOK | dioxane | 1 | 0.5 | 10 | 160 | trace |
| 7 | Mn(CO)$_5$Br | $^t$BuOK | dioxane | 1 | 0.5 | 10 | 160 | trace |
| 8 | [Mn]-III | $^t$BuONa | dioxane | 1 | 0.5 | 10 | 160 | 79 |
| 9 | [Mn]-III | NaOH | dioxane | 1 | 0.5 | 10 | 160 | 83 |
| 10 | [Mn]-III | KOH | dioxane | 1 | 0.5 | 10 | 160 | 67 |
| 11 | [Mn]-III | EtONa | dioxane | 1 | 0.5 | 10 | 160 | 76 |
| 12 | [Mn]-III | Na$_2$CO$_3$ | dioxane | 1 | 0.5 | 10 | 160 | 76 |
| 13 | [Mn]-III | Cs$_2$CO$_3$ | dioxane | 1 | 0.5 | 10 | 160 | 42 |
| 14 | [Mn]-III | Na$_2$CO$_3$ | toluene | 1 | 0.5 | 10 | 160 | 86 |
| 15 | [Mn]-III | Na$_2$CO$_3$ | THF | 1 | 0.5 | 10 | 160 | 50 |
| 16 | [Mn]-III | Na$_2$CO$_3$ | toluene | 1 | 0.5 | 5 | 160 | 86 |
| 17 | [Mn]-III | Na$_2$CO$_3$ | toluene | 1 | 0.5 | 15 | 160 | 86 |
| 18 | [Mn]-III | Na$_2$CO$_3$ | toluene | 1 | 0.2 | 5 | 160 | 93 |
| 19 | [Mn]-III | Na$_2$CO$_3$ | toluene | 1 | 0.1 | 5 | 160 | 76 |
| 20 | [Mn]-III | Na$_2$CO$_3$ | toluene | 1 | 1 | 5 | 160 | 64 |
| 21 | [Mn]-III | Na$_2$CO$_3$ | toluene | 0.5 | 0.2 | 5 | 160 | 76 |
| 22 | [Mn]-III | Na$_2$CO$_3$ | toluene | 2 | 0.2 | 5 | 160 | 93 |
| 23 | [Mn]-III | Na$_2$CO$_3$ | toluene | 1 | 0.2 | 5 | 140 | 50 |
| 24$^a$ | [Mn]-III | Na$_2$CO$_3$ | toluene | 1 | 0.2 | 5 | 160 | 70 |

Reaction conditions: Unless otherwise specified, reactions were performed on a 0.5 mmol scale of $N,N'$-dicyclohexylurea **1a**, 1 mmol 2,3-butanediol **2a**, using 10 mol% of base, 1 mol% of **Mn**-precatalyst, in 0.5 mL dioxane at 160 °C for 16 h. The yields were determined by GC using biphenyl as the internal standard.

$^a$0.75 mmol 2,3-butanediol **2a** was used.

**Fig. 3 | The efficiency of manganese catalyst.**
Manganese-catalyzed dehydrogenative condensation of N,N'-dicyclohexylurea **1a** with 2,3-butanediol **2a** through a gram-scale reaction using 0.05 mol % catalyst loading on a 5 mmol scale of **1a**.

Next, optimization of reaction parameters, including base, solvent, reaction temperature, and the ratio of **1a** to base, was performed using the **[Mn]-III** catalyst. The use of stronger bases, such as $^t$BuONa, NaOH, KOH, and EtONa, afforded **3a** in similar yields (67–83%; Table 1, entries 8–11). The use of a relatively weak base, such as $Cs_2CO_3$, afforded **3a** in 42% yield (Table 1, entry 12). Note that the reaction also proceeded efficiently to afford **3a** in 76% yield when using $Na_2CO_3$ as the base (Table 1, entry 13). Next, the effect of the solvent was investigated using $Na_2CO_3$ as the base. Using THF as the solvent afforded **3a** in a lower yield than that obtained using toluene (**3a** yield: 86%) as the solvent (Table 1, entries 14 and 15). A decrease or increase in the molar percentage of the base (5 and 15 mol%) resulted in similar yields of the desired products (Table 1, entries 16 and 17). Interestingly, the yield of **3a** increased from 86 to 93% on increasing the toluene content from 0.2 to 0.5 mL, keeping the other reaction conditions the same (Table 1, entry 18). The yield of **3a** decreased to a 76% yield on decreasing the toluene content to 0.1 mL (Table 1, entry 19). Contrastingly, increasing the toluene content to 1 mL did not significantly improve the yield of **3a** (Table 1, entry 20). A decrease or increase in the catalyst loading (0.5 and 2 mol%) resulted in lower or same yields of the desired products (Table 1, entries 21 and 22). A decreased yield was obtained when a lower temperature (140 °C) was used (Table 1, entry 23). Notably, the yield of **3a** decreased to 70% when 0.75 mmol **2a** was used as the starting material (Table 1, entry 24). After a systematic investigation considering the maximum utilization rate of the raw material, optimized reaction conditions were identified as 2 equiv. **2a** (based on **1a**) and 1 mol% **[Mn]-III** in the presence of 5 mol% $Na_2CO_3$ in 0.2 mL toluene at 160 °C for 16 h.

**Substrate scope**

Under optimized reaction conditions (Table 1, entry 18), the substrate scope of the reaction was analyzed using **[Mn]-III** as the catalyst. A wide range of substituted 1,2-diols underwent smooth AD condensation reactions with N, N'-dicyclohexylurea, yielding the corresponding imidazolone products in good to excellent yields of 50–99%, as shown in Scheme 1. Notably, imidazolone **3b** was obtained via a reaction between N, N'-dicyclohexylurea and pentane-2,3-diol in 68% yield. The use of the non-substituted 1,2-diol substrate ethylene glycol afforded the corresponding product, imidazolone **3c**, in 86% yield. Next, the substrate applicability of mono-substituted 1,2-diols, which can also react with N, N'-disubstituted ureas, was investigated. For example, methyl-, ethyl-, n-propyl-, isopropyl-, and phenyl-substituted 1,2-diols can react with N, N'-dicyclohexylurea to produce imidazolones **3d**–**3h** in yields of 65–99%. Finally, AD and condensation between N, N'-dicyclohexylurea and cyclic 1,2-diols (1,2-cyclopentanediol and 1,2-cyclohexanediol) were explored, and the corresponding bicyclic imidazolones **3i**-**3j** were obtained in moderate to good yields of 57–85% using 1 mol % **[Mn]-III** as the catalyst.

Reaction conditions: 0.5 mmol scale of N,N'-dicyclohexylurea **1a**, 1 mmol 1,2-diols **2**, using 5 mol% of $Na_2CO_3$, 1 mol% of **[Mn]-III**, in 0.2 mL toluene at 160 °C for 16 h. Isolated yields are shown.

Next, routes for the sustainable synthesis of imidazolones via AD and condensation of ureas with 1,2-diols employing **[Mn]-III** as the catalytic system were investigated. Under optimized reaction conditions, 25 different imidazolones **4a**–**4 y** were synthesized in yields of 42–99% (Scheme 2). A wide range of imidazolones was obtained using ethylene glycol and N, N'-substituted ureas as starting materials. For example, ethyl-, isopropyl-, n-propyl-, and n-butyl-substituted N, N'-dialkylureas afforded the

corresponding imidazolones **4a**–**4 d** in moderate yields of 42–62%. The methyl-, ethyl-, isopropyl-, n-propyl-, n-butyl-, and phenyl-substituted ureas reacted with 2,3-butanediol **2a**, affording imidazolones **4e**–**4j** in moderate to good yields of 53–93%. Furthermore, we attempted the reaction between unsubstituted ureas and diols using this catalytic system. The results showed that monosubstituted urea reacted with the diol to form the target product **4k**, but the yield decreased. However, urea without substituents on either side could not react with diols to obtain the target product in our catalytic reaction system. Moreover, N, N'-dibutyl urea reacted with 1-phenylethane-1,2-diol, affording **4 l** in 93% yield. phenylethane-1,2-diols with halogen (fluorine), ether (methoxy), and heterocycle (pyridine) substituents in the aromatic ring can react with N, N'-dimethylurea to obtain the target products **4m**-**4o** in yields of 79-83%. Additionally, phenylpropane-1,2-diol reacts with N, N'-dimethylurea to obtain **4p** in 88% yield. N, N'-dimethylurea and N, N'-dibutylurea can also react with 1,2-cyclohexanediol, affording **4q** and **4r** in 77 and 99% yield, respectively. Notably, the use of N, N'-dibenzylureas as substrates afforded the corresponding imidazolones **4s**–**4x** in moderate isolated yields of 48–62%. The reaction of 2,3-butanediol with N, N'-dibenzylureas bearing electron-donating substituents, such as methyl and methoxy, afforded the corresponding imidazolone product **4t** and **4 u** in an isolated yield of 54 and 61%. Additionally, N, N'-dibenzylureas bearing electron-withdrawing groups at para positions (p-F, p-Cl or p-CF₃) afforded the corresponding imidazolone products **4 v, 4w** and **4x** in a similar yield, suggesting that electronically biased substituents on aromatic benzyl rings negligibly influence AD and condensation reactions. We performed the reaction in this catalytic system using asymmetric diols and urea. Using 1-ethyl-3-methylurea and 1-phenylpropane-1,2-diol as raw materials, the results showed a mixture of 4 y and 4 y', with a ratio of 1.5:1. Subsequently, we altered the catalyst structure to adjust the product distribution ratio. However, satisfactory results have not been achieved yet. In the future, we will continue to develop different catalytic systems with the aim of changing the product distribution of imidazolones generated by the reaction of asymmetric diols and urea through changes in the catalyst structure or catalytic system.

Reaction conditions: 0.5 mmol scale of N,N'-disubstituted ureas **1**, 1 mmol 1,2-diols **2**, using 5 mol% of $Na_2CO_3$, 1 mol% of **[Mn]-III**, in 0.2 mL toluene at 160 °C for 16 h. Isolated yields are shown.

To further demonstrate the scalability and catalytic efficiency of the proposed reaction and catalyst, respectively, a gram-scale dehydrogenative condensation reaction between N,N'-dicyclohexylurea and 2,3-butanediol was tested using 0.05 mol% catalyst loading on a 5 mmol scale (Fig. 3), and the desired product **3a** was obtained in 83% yield, with the corresponding TON > 1600. Thus, the proposed synthetic route represents a highly practical approach for synthesizing imidazolone derivatives. Moreover, gases in this reaction were collected and analyzed by GC. The GC results indicate that $H_2$ is indeed produced in the reaction system. Therefore, we speculate that dehydrogenation is the reaction mechanism.

**Mechanism experiments**

After the successful Mn-catalyzed dehydrogenative condensation of N, N'-dicyclohexylurea with 2,3-butanediol, a series of experiments was conducted to gain more mechanistic insights. First, poisoning experiments were performed in the presence of $PMe_3$ and $PPh_3$ in sub-stoichiometric amounts with respect to **[Mn]-III** and a drop of Hg (Supporting Table S2). A significant inhibitory effect was not observed in any of the examined cases,

**Fig. 4 | Dehydrogenative condensation with *N*-methyl manganese catalyst.** Comparison of catalytic activity using [**Mn**]-**III** and *N*-methyl-substituted PNP pincer manganese complex [**Mn**]-**V** as the catalyst for initiating the reaction of *N*, *N'*-dicyclohexylurea **1a** with 2,3-butanediol **2a** to afford **3a**.

**Fig. 5 | Control experiments for mechanistic study.** **A** Dehydrogenative condensation of **1a** with 2,3-dimethylbutane-2,3-diol. **B** Dehydrogenative condensation of **1a** with 2-methylbutane-2,3-diol. **C** Dehydrogenative condensation of **1a** with 3-hydroxy-2-oxybutane. **D** Dehydrogenative condensation of **1a** with 2,3-butanedione. **E** [Mn]-IIIA catalyzed dehydrogenative condensation of **1a** with **2a**.

indicating a homogeneous nature of the [**Mn**]-**III** catalyst. Furthermore, *N*-methyl-substituted Mn catalyst [**Mn**]-**V** was used to determine the mechanism behind the dehydrogenative condensation process (Fig. 4). Significantly different catalytic activities are observed when using [**Mn**]-**III** or [**Mn**]-**V** as the catalyst for initiating the reaction of *N*, *N'*-dicyclohexylurea with 2,3-butanediol to afford **3a**, suggesting an outer–sphere mechanism with the metal-ligand cooperativity. The obtained results demonstrate that the outer-sphere dehydrogenation mechanism is the major reaction pathway, and the N–H groups in the [**Mn**]-**III** catalyst are indispensable for the reactions between urea and diols to yield imidazolones or their derivatives.

To further understand the mechanism behind the Mn-catalyzed dehydrogenative condensation reaction of *N*, *N'*-dicyclohexylurea (**1a**) with

2,3-butanediol (**2a**), a set of control experiments were carried out (Fig. 5). First, to demonstrate that the condensation of **2a** occurs via a dehydrogenation process, 2,3-dimethylbutane-2,3-diol **5** and 2-methylbutane-2,3-diol **6** were tested as condensation substrates under optimized conditions, and no products were observed to have formed (Figs. 5A, B). For the reaction in Fig. 5B, we found that in addition to the raw materials, we also detected and isolated 0.45 mmol of the alcohol **6** dehydrogenation product 3-hydroxy-3-methyl-2-butanon **3b**. In contrast, the reaction of 3-hydroxy-2-oxobutane **7** with **1a** occurs smoothly under the optimal reaction conditions or in the absence of the Mn complex and in the absence of both the Mn complex and base (Fig. 5C). The obtained results further support dehydrogenative condensation as the reaction mechanism governing the Mn catalytic system. The generated highly unsaturated Mn complex

**Fig. 6 | Proposed reaction mechanism.** Plausible reaction pathways for acceptorless dehydrogenative condensation of $N, N'$-dicyclohexylurea **1a** with 2,3-butanediol **2a** to synthesis imidazolone **3a**.

**Scheme 1 | Mn-catalyzed dehydrogenative condensation of $N, N'$-dicyclohexylurea with 1,2-diols.**

possibly stabilizes via potential binding with alcohol, yielding an alkoxy–Mn$^I$ complex that subsequently undergoes dehydrogenation under catalytic conditions to produce ketone, releasing $H_2$. The produced ketone then condenses with urea to form the desired product. Moreover, it also indicated that the manganese catalyst only participated in the first dehydrogenation reaction of 2,3-butanediol **2a** to obtain 3-hydroxy-2-oxybutane **7** and did not participate in the subsequent reaction of 3-hydroxy-2-

oxybutane **7** with $N, N'$-dicyclohexylurea **1a** to form **3a**. Using 2,3-butanedione **8** instead of **2a** under optimized conditions or in the absence of Mn catalysts, the reaction did not produce the intended target product **3a** (Fig. 5D). However, we isolated the self-condensation product, 2,5-dimethylhydroquinone, from 2,3-butanedione **8**. We think that 2,3-butanedione **8** is more prone to self-condensation under these conditions, as shown in previous studies[97]. The results indicate that 2,3-butanedione is not formed as

**Scheme 2** | Mn-catalyzed dehydrogenative condensation of ureas with 1,2-diols.

an intermediate product. Non-alkaline reactions catalyzed by only the amido Mn complex [**Mn**]-**IIIA** (Fig. 5E) afforded **3a** in 83% yield, confirming that [**Mn**]-**IIIA** is a catalytically active species for synthesizing imidazolones via Mn-catalyzed dehydrogenation of alcohol. Further mechanistic insights were gained by performing kinetic studies. The kinetic profile of the reaction of **1a** to **2a** is shown in Supplementary Information. It illustrated that there is no induction period for the reaction.

## Proposed mechanism

Based on the results of this study and our previous research[86–89,91], a plausible reaction mechanism behind Mn-catalyzed AD and condensation of N, N'-dicyclohexylurea and 2,3-butanediol to produce imidazolone derivativeshas been proposed (Fig. 6). First, the catalytically active species—amido-manganese complex [**Mn**]-**IIIA** was generated from precatalyst [**Mn**]-**III** in the presence of $Na_2CO_3$. The [**Mn**]-**IIIA** complex further transforms into an

alkoxy–Mn$^I$ complex upon binding with an alcohol substrate. The alkoxy–Mn$^I$ complex undergoes a $\beta$-hydrogen elimination process under catalytic conditions, producing 3-hydroxy-2-oxobutane **7**, releasing a molecule of hydrogen, and generating amino species [**Mn**]-**IIIB**. Subsequently, the 3-hydroxy-2-oxobutane **7** condenses with urea to form an iminium cation **9**, as confirmed by a separate experiment involving a reaction between 3-hydroxy-2-oxobutane **7** and 1,3-dicyclohexylurea (**1a**) under optimized conditions, affording **3a**. The presence of tautomers between iminium cation intermediates **9**, enamine–enol compounds **10**, and carbonyl compounds **11** further promotes the condensation reaction. Then, the equilibrium of the species undergo the second condensation and produce five-membered cyclic iminium intermediates **12** that reorganize to enamines to form the final imidazolone derivatives **3a**. Furthermore, the release of H$_2$ from the [**Mn**]-**IIIB** species regenerates the metal catalyst for the next catalytic cycle.

An effective Mn pincer-catalyzed AD and condensation process for synthesizing imidazolones starting from urea and diols using a highly selective base, namely Na$_2$CO$_3$, has been reported. It was a successful utilization of a non-noble metal catalyst for synthesizing imidazolones. The proposed catalytic reaction exhibited excellent functional group tolerance. The reaction to produce imidazolones proceeded smoothly in the presence of various functional groups in urea or diol substrates, yielding the corresponding products (>30 cases examined) in high yields up to 99%. The desired reaction proceeded even at a low catalyst loading of 0.05 mol%, with a high TON of 1660. Mechanistic investigations were performed using several control experiments to identify the key reaction intermediates in the target catalytic system, and a plausible mechanistic pathway has been proposed. Considering the easy availability of non-noble metals and biomass-derived substrates, the proposed catalytic reaction for synthesizing imidazolones can be anticipated to complement the current methods in organic synthesis.

## Methods

### General procedure for Mn-catalyzed dehydrogenative condensation of N,N'-disubstituted ureas with 1,2-diols

All experiments were carried out in a 15 mL pressure seal tube. In the argon atmosphere glovebox, N, N'-disubstituted ureas **1** (0.5 mmol), 1,2-diols **2** (1.0 mmol), [**Mn**]-**III** (1 mol%), Na$_2$CO$_3$ (5 mol%) were added sequentially to the seal tube equipped with a magnetic stir bar. The reaction mixture was stirred at 160 °C for 16 h and cooled to room temperature. After the gas was released, the resulting solution was concentrated in a vacuum, and the residue was purified by chromatography on silica gel, eluting with the mixture of ethyl acetate/petroleum ether to give the corresponding imidazolone products.

## Data availability

The authors declare that all the data supporting the findings of this research are available within the article and its supplementary information.

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

## Acknowledgements

We acknowledge the financial support from the National Natural Science Foundation of China (22302056), China Tobacco Fund of Hebei (No. HBZY2024A046) and the top-notch personnel fund of Henan agricultural university (Grant 30501028).

## Author contributions

Z.S. and M.Z. conceived and designed the experiments. J.D., D.K., T.K., S.L., and S.S. performed the experiments and analyzed the data. B.C. participated in the discussions. Z.S. supervised the research and wrote the manuscript with the assistance of other authors. All authors interpreted the results in the manuscript.

## Competing interests

The authors declare no competing interests.
