## [Transparent Peer Review file · Communications Chemistry]

Mn-Catalysed Acceptorless Dehydrogenative Condensation of Ureas with 1,2-Diols for Synthesizing Imidazolones

Corresponding Author: Dr Zihui Shao

Version 0:

Reviewer comments:

Reviewer #1

(Remarks to the Author)

Zhao, Shao and co-workers report here an interesting work in which they apply for the first time a non-noble homogeneous based catalytic system for the synthesis of imidazolones from ureas and diols. The catalytic activity of Mn pincer complexes, such as the ones reported in this work, in acceptorless dehydrogenative condensations or borrowing hydrogen processes is not new, and neither it is new the synthetic approach for obtaining imidazolones through ADC from ureas and diols. Despite this lack of novelty, demonstrating such an interesting approach for an underdeveloped synthesis of heterocycles is valuable and from the point of view of this referee deserves the publication of this work in a general interest journal such as Communications Chemistry. However, in my opinion certain points should be improved before its publication:

- In general writing should be improved in some points. Specifically in the first paragraph of introduction in which biological activities are discussed. In my opinion, nitrogen containing heterocyclics or N-heterocyclics is not correct, it should be N-heterocycles.
- In Fig. 2: Capital letters should be used at the beginning of the titles, e.g. A) heterogeneous; in the caption should be Pd instead Pa; references or the name of corresponding authors should appear in the figure.
- Substrate scope should be improved including more functionalities such as halogens, ethers, heterocycles in both the urea or the alcohol. It would be interesting to explore the reactivity of phenylethane-1,2-diols and phenylpropane-1,2-diols with different substitution in the aromatic ring, as well as N,N-dipheylureas with substitutions.
- Mechanistic studies need to be expanded from several viewpoints:
 - o Kinetic studies are missing and they can give a very valuable information regarding the possibility of an induction period, the calculation of reaction orders. It would be very interesting to perform kinetics and compare initial rate using keto alcohol 7 and urea 1a as starting materials.
 - o Very related with my last comment, reaction between 1a and 7 (Fig. 5D) should be performed in the absence of the Mn complex and in the absence of the Mn complex and the base. If you observe your mechanism the process should work without catalyst from 7.
 - o Page 12, line 182 "Comparable catalytic activities are observed when using [Mn]-III or [Mn]-V" I would not say comparable catalytic activities, it would be more very different or am I missing something?
 - o Is any product observed in reaction showed in Fig5B? Perhaps an open product?
 - o H₂ detection would be a strong evidence of the mechanism.

Reviewer #2

(Remarks to the Author)

This manuscript reports a manganese-catalyzed acceptorless dehydrogenative condensation (ADC) of N,N'-disubstituted ureas with 1,2-diols for the synthesis of imidazolones. The work presents a notable advance by employing an earth-abundant, non-noble metal (Mn) catalyst providing access to a range of imidazolone derivatives with high yield and functional group tolerance. The topic is timely and relevant to green chemistry, particularly in the context of heterocycle synthesis. However, several points require clarification before consideration for acceptance:

1. Why do the authors choose 4:1 stoichiometry of diol:urea whereas the product needs 1:1? What is happening with the remaining diol?
2. Table 1, entries 17-19, how do the authors rationalise the effect of toluene on reaction yield? Such a sharp effect on yield by varying the amount of toluene is surprising.

3. In the proposed mechanism section the authors say - "First, the [Mn]-III catalyst releases a hydrogen bromide molecule upon reaction with Na₂CO₃ while producing a catalytically active species—amino-manganese complex [Mn]-III A."

What is the evidence of the formation of HBr? It is likely that the reaction would produce NaBr, H₂O, and CO₂. Additionally, complex [Mn]-III A is an amido-species not an amino-species. Complex [Mn]-III B is an amino-species. These should be corrected.

4. Why does the reaction not work with 2,3-butanedione (8, Figure 5)? Why is the release of H₂ necessary for the reported transformation?

5. In cases where asymmetric diols or ureas are used, was any regio- or diastereoselectivity observed or measured? How does the catalyst influence product distribution in such cases?

6. How does the reported manganese catalyst compare with previously reported ruthenium catalyst for this transformation?

Reviewer #3

(Remarks to the Author)

This study presents a manganese-catalyzed, dehydrogenative condensation of ureas with 1,2-diols to synthesize imidazolones. Imidazolones are useful motifs, which can be synthesised by Pd or Ru catalysis, but this paper presents an alternative approach with Mn catalysts. The paper shows some reaction optimization, followed by scope study with various symmetric ureas and diols. All examples seem to work well. A gram-scale reaction is also given. Some mechanistic investigations and a proposed mechanism are given (notes below on this).

The reaction presented is very interesting and could be of good use, benefitting from the use of earth abundant Mn metal. In general the manuscript is well-presented, has a clear logical structure and is easy to read. I can see this manuscript being published and well-received. However, there are a few queries and issues that should be addressed before publication as follows:

One limitation is the omission of discussion on stereochemistry. Three of the four medicinal chemistry examples given in introduction are non-symmetric imidazolones. What happens in the proposed reaction if starting from ureas with different substituents on each side in combination with diols with different R groups. Is there some regiocontrol based on sterics or electronics. It would be nice to acknowledge this question and give some examples of reaction outcome. Do unsubstituted ureas work? i.e., NH₂CONH₂? Or RNHCONH₂? That might be a more useful starting point from an application point of view.

Fig 5 control experiments. You say often that "no product 3A" forms, but it might be more interesting to say what does happen. For example, reaction D with the di-ketone...does the urea react with the di-ketone, but then 'get stuck' in an oxidised form of 3A. i.e., the initial dehydrogenation of diol is needed, so that hydrogenation can occur from the proposed 'Mn-H₂'-like species later on in the mechanism. Please be clear (or investigate) what does happen in each of these 'failed' reactions.

Fig 5 E brings in a new catalyst [Mn]-III A. Where does that come from? It doesn't seem to be discussed/synthesized in the supporting information.

Overall, I find the mechanistic study a little simplistic. More work and a more appropriate, detailed mechanism is needed in my opinion. Specifically:

- First big question is that you have an arrow with H₂ leaving the system from a Mn intermediate. What is triggering that release step? Seems unusual just to have H₂ spontaneously leaving the Mn – N intermediate. Does the reaction show bubbling H₂ gas? For example, in the gram scale reaction you'd be making... 20 cm³ of gas. Can you provide evidence for loss of H₂?

- According to your mechanism, once we have compound 7 (3-hydroxy-2-oxobutane) the reagents come together to make product 3A (without any further need for catalyst or base). However, in Fig 5 C, you react 7 with catalyst and base to get to product 3A. If your mechanism is correct, you should be able to react 7 with urea to make product 3A without any additional reagents. ...or if not, the mechanism needs re-thinking.

- The step in the mechanism that seems too simplistic is the step starting from the intermediate on the left hand side of the square bracket. {note here that numbering these intermediates would be helpful}. It's not at all clear how you get from this C(OH) + NH to a C=N+ bond. That step at first glance looks more like it's set up for another dehydrogenation to make the ketone, before reaction with urea N, followed by hydrogenation to get the final aromatic product. In the style of the classic 'borrowing hydrogen' mechanism. Or more plausible perhaps is the hydrogenation of the C=N+ already present at this stage by the Mn-N-H₂ intermediate. Followed by another dehydrogenation of the remaining CH-OH...but then we still have the catalyst left as Mn-N-H₂, which takes us back to point 1 above.

Some minor comments on phrasing:

- P3 Line 26. Saying "numerous natural products" is a bit vague. Can you give a couple of specific examples.
- P3 line 40 "However, traditional synthetic methods cannot satisfy the increasing demand for imidazolones.". I don't understand this statement. Why cant all these methods provided 'satisfy the increasing demand' ?
- Line 43 "Alcohol is highly abundant..." it sounds like you're talking about ethanol (or vodka!). Rephrase
- Line 51 "Pa-catalyzed" typo!
- Line 60: "The previously reported catalytic systems require advanced specialized materials ..." sounds too vague. Be clear what the 'advanced specialized materials' are that you are concerned about.

Version 1:

Reviewer comments:

Reviewer #1

(Remarks to the Author)

The authors successfully responded to all the points and in my opinion the manuscript should be published. I would encourage the authors to include some of the new experiences performed for responding to the referees (those not included, such as hydrogen detection) in the paper, at least in the supporting information.

Reviewer #2

(Remarks to the Author)

The authors have conducted all the necessary revisions satisfactorily. I am happy for the paper to be now accepted to the Communication Chemistry.

Reviewer #3

(Remarks to the Author)

Overall, it appears all the corrections required have been completed. Some issues to consider below before publication:

Figure 4. Yield is down to 8% when N-Me catalyst used. But why does 8% product still form when the NH is apparently crucial for reaction to proceed. Does this suggest there's another mechanism at work here?

In the mechanism text referring to Fig 5,

- Need to bold the number 6 in line 8
- Typo on line 9. 3-hydroxy-3-me....3c should be 3b

In experimental section in Supp Info, need to include mass, % yield of the imidazole products. At the moment its just NMR data.

I've started to take a detailed look at the NMR spectra in the Supporting Info and have some concerns.

- Fig 13: Looks OK
- Fig 15 doesn't look pure. There are clearly multiplet peaks around 2ppm not assigned. Also we can see the ppm range used for integrals raises concern. For the peak at 1.4ppm (integral 6.20) the ppm range doesn't look to include the full peak (more of this later on)
- Fig 17: good
- Fig 19 good, but again little impurity at about 1.9ppm multiplet
- Fig 21: again large peak at 2.2ppm not accounted for. Also, compare the 6.12 integral at 1.7ppm and the 2.19 integral at 2.5ppm. By eye, the former is not 3X greater integral than latter. And looking at the narrowness of integral range on 2.5ppm peak suggests not all peak is included in integral reported. By contrast the ppm range on 1.7ppm peak has been exaggerated to include neighbouring signal.

I've scanned down later ones and they generally look OK, although some impurity peaks on occasion.

- Fig 68 look concerning again. Multiplet at 1.1ppm (4.37 integral) is clearly not integrating full peak. Compare to broad singlets at 2.2ppm (4.01 integral) and 1.5ppm (4.03 integral). By eye, the two latter signals are clearly lower integral than the former multiplet at 1.1ppm, despite both being labelled as having integral around 4.

My overall conclusion is that the reaction does work, but the authors have had some poor purification and chosen to include impure material in yield calculation. With some manipulation of integral values to match the expected numbers. It's important not to cut these corners and we need to either see the raw data files of the spectra picked out here or the authors go through thoroughly, repurify where necessary and clearly integrate the full NMR peaks.

河南农业大学

中国 郑州 450002

Henan Agricultural University

Zhengzhou 450002, China

Dr. Zhihui Shao, *Associate Professor*

College of Tobacco Science, Henan Agricultural University

Zhengzhou 450002, P. R. China

shaozh21@henau.edu.cn

August 8th, 2025

Dear Editor:

Thank you very much for handling our manuscript titled “Mn-Catalysed Acceptorless Dehydrogenative Condensation of Ureas with 1,2-Diols for Synthesizing Imidazolones” (Manuscript ID: COMMSCHEM-25-0358-T). We do appreciate the comments and suggestions from all the reviewers. Accordingly, we carefully revised this manuscript to address most of the comments and questions from the reviewers. Please find our statements and answers to all the suggestions and questions point by point as follows:

COMMENTS TO AUTHOR:

Reviewer: #1

Comments:

Zhao, Shao and co-workers report here an interesting work in which they apply for the first time a non-noble homogeneous based catalytic system for the synthesis of imidazolones from ureas and diols. The catalytic activity of Mn pincer complexes, such as the ones reported in this work, in acceptorless dehydrogenative condensations or borrowing hydrogen processes is not new, and neither it is new the synthetic approach for obtaining imidazolones through ADC from ureas and diols. Despite this lack of novelty, demonstrating such an interesting approach for an underdeveloped synthesis of heterocycles is valuable and from the point of view of this referee

deserves the publication of this work in a general interest journal such as Communications Chemistry. However, in my opinion certain points should be improved before its publication:

Concerns:

In general writing should be improved in some points. Specifically in the first paragraph of introduction in which biological activities are discussed. In my opinion, nitrogen containing heterocyclics or N-heterocyclics is not correct, it should be N-heterocycles.

Our response: Thank you for your comments and suggestions. We have modified “nitrogen containing heterocyclics or N-heterocyclics” to “N-heterocycles” in the revised manuscript with track changes. In addition, we have revised the discussion on biological activities in the first paragraph of the Introduction. The revised content is as follows: Among N-heterocycles, imidazolones and their derivatives hold significant value in the synthesis of natural products, such as dibromophakelstatin and axinohydantoin. They are integral components of several currently marketed drugs, largely because of their notable antiprotozoal and antiviral properties. For example, loxoribine, a guanosine analog, is known for its antiviral and antitumor activities. It functions as a synthetic adjuvant for antitumor responses. Another example is theacrine, a methylpurine derivative that strengthens the heart, promotes diuresis, dilates the coronary arteries, relaxes bronchial smooth muscle, and stimulates the central nervous system. It is primarily used to treat bronchial asthma, emphysema, bronchitis, and cardiac dyspnea. Furthermore, imidazolone units form the core structures of important drug intermediates such as Catramilast and Metazamide. These revisions have been made in the manuscript with tracked changes.

In Fig. 2: Capital letters should be used at the beginning of the titles, e.g. A) heterogeneous; in the caption should be Pd instead Pa; references or the name of corresponding authors should appear in the figure.

Our response: Thank you for your comments and suggestions. We have modified “heterogeneous” to “Heterogeneous” and “Pa” to “Pd” in the revised manuscript with track

changes. Meanwhile, the references and the names of corresponding authors have been added to the figure.

Substrate scope should be improved including more functionalities such as halogens, ethers, heterocycles in both the urea or the alcohol. It would be interesting to explore the reactivity of phenylethane-1,2-diols and phenylpropane-1,2-diols with different substitution in the aromatic ring, as well as *N,N*-dipheylureas with substitutions.

Our response: Thank you for your comments and suggestions. In accordance with the reviewer's suggestions, we have expanded the functional group range of the substrate in the revised manuscript. For example, phenylethane-1,2-diols with halogen (fluorine), ether (methoxy), and heterocycle (pyridine) substituents in the aromatic ring can react with urea to obtain the target products **4m-4o** in yields of 79-83%. Additionally, phenylpropane-1,2-diol and *N,N*-diphenylurea can also react with urea and 1,2-diols respectively to obtain the target products **4j** and **4p** in 53% and 88% yields, respectively, indicating that the reaction system has good functional group compatibility.

Mechanistic studies need to be expanded from several viewpoints:

Kinetic studies are missing and they can give a very valuable information regarding the possibility of an induction period, the calculation of reaction orders. It would be very interesting to perform kinetics and compare initial rate using keto alcohol **7** and urea **1a** as starting materials.

Our response: Thank you for your comments and suggestions. By monitoring the product concentration under the optimal reaction conditions at different time points, we found that the product concentration was directly proportional to time (**Fig. R1**). This indicates that within the studied time range, the reaction rate was constant and there was no obvious induction period. This result indicates that the reaction proceeds at a stable rate from the beginning without a distinct induction stage. To confirm this, we repeated the experiment using different initial solvent amounts. The results show that when the initial solvent volume was 1 mL, the

relationship between the product concentration and time remained linear (**Fig. R2**). This further supports the conclusion that there is no obvious induction period for the reaction. Because the product concentration is directly proportional to time and this relationship still holds under different initial solvent amounts, we initially believe that this reaction may be a zero-order reaction. In other words, the reaction rate is independent of the reactant concentration and is only related to time.

Fig. R1

Fig. R2

Very related with my last comment, reaction between 1a and 7 (Fig. 5C) should be performed in the absence of the Mn complex and in the absence of the Mn complex and the base. If you observe your mechanism the process should work without catalyst from 7.

Our response: Thank you for this suggestion. We performed the reaction between 1a and 7 in the absence of the Mn complex and in the absence of both the Mn complex and base. The results

show that the reactions of **1a** and **7** occur smoothly in the absence of the Mn complex and in the absence of both the Mn complex and base. The corresponding results have been added to **Fig. 5C** in the revised manuscript.

Page 12, line 182 “Comparable catalytic activities are observed when using [Mn]-III or [Mn]-V” I would not say comparable catalytic activities, it would be more very different or am I missing something?

Our response: Thank you for this suggestion. We have modified this description in the manuscript with track changes. The revised expression is as follows: Significantly different catalytic activities are observed when using [Mn]-III or [Mn]-V as the catalyst for initiating the reaction of *N,N'*-dicyclohexylurea with 2,3-butanediol to afford **3a**.

Is any product observed in reaction showed in Fig 5B? Perhaps an open product?

Our response: Thank you for this suggestion. In the control experiment, we conducted a detailed study on the failed reaction products. For the reaction in **Fig. 5B**, through GC-MS detection, we found that in addition to the raw materials, we also detected and isolated 0.45 mmol of the alcohol dehydrogenation product 3-hydroxy-3-methyl-2-butanone **3b**. For the reaction shown in **Fig. 5D**, the same results were obtained by GC-MS in the presence and absence of the Mn catalyst. Only a small amount of the target product was generated. We also isolated the self-condensation product, 2,5-dimethylhydroquinone **3c**, from 2,3-butanedione **8**. We think that 2,3-butanedione **8** is more prone to self-condensation under these conditions, as shown in previous studies (*Org. Biomol. Chem.*, **2022**, *20*, 6445–6458).

H₂ detection would be a strong evidence of the mechanism.

Our response: Thanks for the suggestion from this reviewer. Gases in the gram-scale reaction after the reaction of **1a** with **2a** (**Fig. 3**) were collected and analyzed by GC. As shown in the figure below, the evolved gas was released from the system, collected, and measured using the

following apparatus. The results of the GC analysis are presented below.

Gas collector

GC analysis of the gas phase

GC conditions: Packed Column. Inlets: 100 °C; Detector: BID 200 °C; Carrier Gas: He; Flow: 51.4 mL/min; Oven: 35 °C, hold 2 min; 5 °C /min to 80 °C, hold 5 min.

The GC results shown in the above figure indicate that H₂ is indeed produced in the reaction system. Therefore, we speculate that dehydrogenation is the reaction mechanism.

Reviewer: #2

Comments:

This manuscript reports a manganese-catalyzed acceptorless dehydrogenative condensation (ADC) of N,N'-disubstituted ureas with 1,2-diols for the synthesis of imidazolones. The work presents a notable advance by employing an earth-abundant, non-noble metal (Mn) catalyst providing access to a range of imidazolone derivatives with high yield and functional group tolerance. The topic is timely and relevant to green chemistry, particularly in the context of heterocycle synthesis. However, several points require clarification before consideration for acceptance:

Concerns:

1. Why do the authors choose 4:1 stoichiometry of diol:urea whereas the product needs 1:1? What is happening with the remaining diol?

Our response: Thank you for your comments. The optimal diol to urea ratio (2:1) was determined by detailed optimization of the conditions. After the reaction was completed, in addition to the target product **3a**, we separated the product 2,5-dimethylhydroquinone **3c**, formed by the self-condensation of 2,3-butanedione **8** after the complete dehydrogenation of 2,3-butanediol. We think that 2,3-butanedione **8** is more prone to self-condensation under these conditions, as shown in previous studies (*Org. Biomol. Chem.*, **2022**, *20*, 6445-6458). Therefore, this reaction requires a relatively large amount of the diol to obtain the target product in high yield.

2. Table 1, entries 17-19, how do the authors rationalise the effect of toluene on reaction yield? Such a sharp effect on yield by varying the amount of toluene is surprising.

Our response: Thank you for your comments. During the experiment, the solubility of urea in toluene was poor. When the amount of toluene was 0.1 mL, urea could not be completely dissolved in toluene, which would prevent urea from reacting with diols in a timely manner, thus decreasing the yield of the target product. However, when the amount of solvent was too

high (1 mL), although the raw material could be completely dissolved, the complete dehydrogenation product of 2,3-butanediol, 2,3-butanedione **8**, partially underwent self-condensation reactions to form the byproduct 2,5-dimethylhydroquinone **3c**, as shown in previous studies (*Org. Biomol. Chem.*, **2022**, *20*, 6445-6458), thereby reducing the yield of the target product **3a**. In fact, we detected 0.11 mmol of **3c** generation in Table 1 Entry 16, and 0.15 millimoles of **3c** generation in Table 1 Entry 18. However, we detected 0.24 mmol of **3c** generation in Table 1 Entry 20.

3. In the proposed mechanism section the authors say - "First, the [Mn]-III catalyst releases a hydrogen bromide molecule upon reaction with Na₂CO₃ while producing a catalytically active species—amino-manganese complex [Mn]-IIIA."

What is the evidence of the formation of HBr? It is likely that the reaction would produce NaBr, H₂O, and CO₂. Additionally, complex [Mn]-IIIA is an amido-species not an amino-species. Complex [Mn]-IIIB is an amino-species. These should be corrected.

Our response: Thank you for your reminder. In our previous study (*J. Am. Chem. Soc.* **2017**, *139*, 11941-11948), we conducted a detailed investigation of the reactivity of ⁱPrPNP complex [Mn]-I. We found that [Mn]-I could be converted into the Mn(I) PNP amido complex [Mn]-IA in the presence of a base. This is an acid-base reaction. As the reviewers pointed out, it is very likely to produce NaBr, H₂O and CO₂ instead of HBr. Based on this, we have modified the corresponding expression "the [Mn]-III catalyst releases a hydrogen bromide molecule upon reaction with Na₂CO₃ while producing a catalytically active species—amido-manganese complex [Mn]-IIIA" to "the catalytically active species—amido-manganese complex [Mn]-IIIA was generated from precatalyst [Mn]-III in the presence of Na₂CO₃" in the revised manuscript with track changes. Furthermore, we have modified the corresponding [Mn]-IIIA and [Mn]-IIIB to include amido and amino species in the main text.

4. Why does the reaction not work with 2,3-butanedione (**8**, Figure 5)? Why is the release of H₂ necessary for the reported transformation?

Our response: Thank you for your comment. In the controlled experiment of **Fig. 5D**, we conducted a detailed study of the reaction products. The same results were obtained by GC-MS in the presence and absence of the **Mn** catalyst. Only a small amount of the target product was generated. We also isolated the self-condensation product, 2,5-dimethylhydroquinone **3c**, from 2,3-butanedione **8**. We think that 2,3-butanedione **8** is more prone to self-condensation under these conditions, as shown in previous studies (*Org. Biomol. Chem.*, **2022**, *20*, 6445–6458). In addition, the reaction requires the first dehydrogenation of 1,2-diols catalyzed by a Mn catalyst to obtain 3-hydroxy-2-oxobutane **7**. Based on our previous study on Mn-catalyzed alcohol dehydrogenation (*J. Am. Chem. Soc.* **2017**, *139*, 11941–11948), the amino species **[Mn]-IIIB** obtained from the dehydrogenation of the alcohol needs to release H₂ to obtain the catalytically active species—amido-manganese complex **[Mn]-IIIA** and continue to participate in the subsequent dehydrogenation process. For further verification, gases in the gram-scale reaction after the reaction of **1a** with **2a** (**Fig. 3**) were collected and analyzed by GC. As shown in the figure below, the evolved gas was released from the system, collected, and measured using the following apparatus. The results of the GC analysis are presented below.

Gas collector

GC analysis of the gas phase

GC conditions: Packed Column. Inlets: 100 °C; Detector: BID 200 °C; Carrier Gas: He; Flow: 51.4 mL/min; Oven: 35 °C, hold 2 min; 5 °C /min to 80 °C, hold 5 min.

The GC results shown in the above figure indicate that H₂ is indeed produced in the reaction system. Therefore, we speculate that dehydrogenation is the reaction mechanism.

5. In cases where asymmetric diols or ureas are used, was any regio- or diastereoselectivity observed or measured? How does the catalyst influence product distribution in such cases?

Our response: Thank you for your comments and suggestions. We performed the reaction in this catalytic system using asymmetric diols and urea. First, using 1-ethyl-3-methylurea and 1-phenylpropane-1,2-diol as raw materials, after the reaction, GC-MS detection revealed the formation of two products with molecular weights consistent with those of the target products. Column separation achieved an 83% yield. The NMR spectrum results showed a mixture of 4y and 4y', with a ratio of 1.5:1. The HRMS results indicated that the molecular weights were consistent with those of the target product. The relevant results have been included in the revised manuscript. In addition, we attempted a reaction using 1-ethyl-3-methylurea and pentane-2,3-diol as raw materials in this catalytic system. The GC-MS results showed the same reaction results as those mentioned above. Subsequently, we altered the catalyst structure to adjust the product distribution ratio. However, satisfactory results have not been achieved yet. In the future, we will continue to develop different catalytic systems with the aim of changing the product distribution of imidazolones generated by the reaction of asymmetric diols and urea through changes in the catalyst structure or catalytic system.

GC-MS spectrum

GC-MS spectrum

ShaoZhiHui_238.nd
szh-jq-23-5
H1
CD3CN
238/ShaoZhiHui
PROTON CD3CN F:\Administrator 26

NMR spectrum

HRMS spectrum

GC-MS spectrum

GC-MS spectrum

6. How does the reported manganese catalyst compare with previously reported ruthenium catalyst for this transformation?

Our response: Thank you for your comment. In 1992, Watanabe *et al.* reported a homogeneous $\text{RuCl}_2(\text{PPh}_3)_3$ -catalyzed synthesis of imidazolones from N,N' -disubstituted ureas and 1,2-diols. Herein, we report the first catalytic system based on non-noble metal catalysis using well-defined manganese (Mn) catalysts. Comparing the pioneering work of Watanabe *et al.* to this study, a lower reaction temperature (160 vs. 180 °C) is required for the condensation of ureas with 1,2-diols described here. Additionally, the catalyst loading for the condensation reaction in the Mn system reported herein was lower than that required for the previously reported Ru system (1 mol% vs. 4 mol%). Moreover, in these two studies, different catalytic systems were used to perform the same reactions using the same raw materials. For instance, in the substrate expansion, the products **4e** and **4q** were synthesized in both cases. The manganese catalytic system was able to obtain the desired products in higher yields (**4e**: 86% vs. 73% yield; **4q**: 77% vs. 51% yield) at a lower temperature and with a reduced amount of the catalyst. Therefore, the efficiency of the newly proposed catalytic system (based on Mn catalysis) was comparable to that of the previously reported Ru system.

Reviewer: #3

Comments:

This study presents a manganese-catalyzed, dehydrogenative condensation of ureas with 1,2-diols to synthesize imidazolones. Imidazolones are useful motifs, which can be synthesised by Pd or Ru catalysis, but this paper presents an alternative approach with Mn catalysts. The paper shows some reaction optimization, followed by scope study with various symmetric ureas and diols. All examples seem to work well. A gram-scale reaction is also given. Some mechanistic investigations and a proposed mechanism are given (notes below on this).

The reaction presented is very interesting and could be of good use, benefitting from the use of earth abundant Mn metal. In general the manuscript is well-presented, has a clear logical structure and is easy to read. I can see this manuscript being published and well-received. However, there are a few queries and issues that should be addressed before publication as follows:

Concerns:

One limitation is the omission of discussion on stereochemistry. Three of the four medicinal chemistry examples given in introduction are non-symmetric imidazolones. What happens in the proposed reaction if starting from ureas with different substituents on each side in combination with diols with different R groups. Is there some regiocontrol based on sterics or electronics. It would be nice to acknowledge this question and give some examples of reaction outcome. Do unsubstituted ureas work? i.e., NH_2CONH_2 ? Or RNHCONH_2 ? That might be a more useful starting point from an application point of view.

Our response: Thank you for your comment. We performed the reaction in this catalytic system using asymmetric diols and urea. First, using 1-ethyl-3-methylurea and 1-phenylpropane-1,2-diol as raw materials, after the reaction, GC-MS detection revealed the formation of two products with molecular weights consistent with those of the target products. Column separation achieved an 83% yield. The NMR spectrum results showed a mixture of 4y and 4y', with a ratio of 1.5:1. The HRMS results indicated that the molecular weights were consistent

with those of the target product. The relevant results have been included in the revised manuscript. In addition, we attempted a reaction using 1-ethyl-3-methylurea and pentane-2,3-diol as raw materials in this catalytic system. The GC-MS results showed the same reaction results as those mentioned above. Subsequently, we altered the catalyst structure to adjust the product distribution ratio. However, satisfactory results have not been achieved yet. In the future, we will continue to develop different catalytic systems with the aim of changing the product distribution of imidazolones generated by the reaction of asymmetric diols and urea through changes in the catalyst structure or catalytic system. Furthermore, we attempted the reaction between unsubstituted ureas and diols using this catalytic system. The results showed that monosubstituted urea reacted with the diol to form the target product **4k**, but the yield decreased. However, urea without substituents on either side could not react with diols to obtain the target product in our catalytic reaction system.

GC-MS spectrum

GC-MS spectrum

ShaoZhiHui.238.nd
szh-jq-23-5
H1
CD3CN
238/ShaoZhiHui
PROTON CD3CN F:\Administrator 26

3.73, 3.71, 3.69, 3.67, 3.65, 3.61, 3.60, 3.58, 3.20, 3.08, 1.24, 1.22, 1.21, 0.99, 0.97, 0.95

NMR spectrum

HRMS spectrum

GC-MS spectrum

GC-MS spectrum

Fig 5 control experiments. You say often that “no product 3A” forms, but it might be more interesting to say what does happen. For example, reaction D with the di-ketone... does the urea react with the di-ketone, but then ‘get stuck’ in an oxidised form of 3A. i.e., the initial dehydrogenation of diol is needed, so that hydrogenation can occur from the proposed ‘Mn-H2’-like species later on in the mechanism. Please be clear (or investigate) what does happen in each of these ‘failed’ reactions.

Our response: Thanks for the suggestion from this reviewer. In the control experiment, we conducted a detailed study of the failed reaction products. For the reaction in **Fig. 5B**, through GC-MS detection, we found that in addition to the raw materials, we also detected and isolated 0.45 mmol of the alcohol dehydrogenation product 3-hydroxy-3-methyl-2-butanone **3b**. For the

reaction shown in **Fig. 5D**, the same results were obtained by GC-MS in the presence and absence of the **Mn** catalyst. Only a small amount of the target product was generated. We also isolated the self-condensation product, 2,5-dimethylhydroquinone **3c**, from 2,3-butanedione **8**. We think that 2,3-butanedione **8** is more prone to self-condensation under these conditions, as shown in previous studies (*Org. Biomol. Chem.*, **2022**, *20*, 6445–6458). Furthermore, in the control experiment shown in **Fig. 5C**, 3-hydroxy-2-oxybutane **7** and **1a** reacted to afford **3a** in the absence of the Mn complex and base, with >99% yield. Therefore, we believe that the manganese catalyst only participated in the first dehydrogenation reaction of 2,3-butanediol **2a** to obtain 3-hydroxy-2-oxybutane **7** and did not participate in the subsequent reaction of 3-hydroxy-2-oxybutane **7** with *N,N'*-dicyclohexylurea **1a** to form **3a**.

Overall, I find the mechanistic study a little simplistic. More work and a more appropriate, detailed mechanism is needed in my opinion. Specifically:

- First big question is that you have an arrow with H₂ leaving the system from a Mn intermediate. What is triggering that release step? Seems unusual just to have H₂ spontaneously leaving the Mn–N intermediate. Does the reaction show bubbling H₂ gas? For example, in the gram scale reaction you'd be making... 20 cm³ of gas. Can you provide evidence for loss of H₂?

Our Response: Thanks for the suggestion from this reviewer. In our previous study (*J. Am. Chem. Soc.* **2017**, *139*, 11941–11948), we conducted a detailed study on the reactivity of the ⁱPrPNP complex **[Mn]-I**. First, the complex **[Mn]-I** could be converted into the Mn(I) PNP amido complex **A** in the presence of 3 equiv of base with 69% isolated yield. Then, amido complex **A** was reacted with H₂ (10 bar) at room temperature to generate the cis-hydride complex **B** in full conversion. Heating a toluene solution of **B** at 30 °C for 4 h led to the formation of a mixture of **B**, **B'**, and **A** in a ratio of 1:0.21:0.36.

Based on our previous research results, a similar mechanism is proposed in this work. However, to further illustrate that H₂ was generated in the reaction system, gases in the gram-scale reaction after the reaction of **1a** with **2a** (Fig. 3) were collected and analyzed by GC. As shown in the figure below, the evolved gas was released from the system, collected, and measured using the following apparatus. The results of the GC analysis are presented below. (The yield of H₂ was calculated based on the maximum H₂ evolution with respect to 100% conversion of **1a** to **3a**. The molar volume of hydrogen at 20 °C and 1 atm pressure was taken as 24.1 L.)

GC analysis of the gas phase

GC conditions: Packed Column. Inlets: 100 °C; Detector: BID 200 °C; Carrier Gas: He; Flow: 51.4 mL/min; Oven: 35 °C, hold 2 min; 5 °C /min to 80 °C, hold 5 min.

The GC results shown in the above figure indicate that 88 mL of H₂ is indeed produced in the reaction system. Therefore, we speculate that dehydrogenation is the reaction mechanism. The calculation results showed that the yield of generated hydrogen was as follows:

$$\text{Yield of H}_2 \text{ for the reaction} = 88 / (24.1 * 5 * 0.83) = 87.9\%$$

- According to your mechanism, once we have compound 7 (3-hydroxy-2-oxobutane) the reagents come together to make product 3A (without any further need for catalyst or base). However, in Fig 5C, you react 7 with catalyst and base to get to product 3A. If your mechanism is correct, you should be able to react 7 with urea to make product 3A without any additional reagents. ...or if not, the mechanism needs re-thinking.

Our response: Thank you for your comments and suggestions. We performed the reaction between **1a** and **7** in the absence of the Mn complex and in the absence of both the Mn complex and base. The results show that the reactions of **1a** and **7** occur smoothly in the absence of the Mn complex and in the absence of both the Mn complex and base. The corresponding results have been added to **Fig. 5C** in the revised manuscript.

The step in the mechanism that seems too simplistic is the step starting from the intermediate on the left hand side of the square bracket. {note here that numbering these intermediates would be helpful}. It's not at all clear how you get from this C(OH) + NH to a C=N⁺ bond. That step at first glance looks more like it's set up for another dehydrogenation to make the ketone, before reaction with urea N, followed by hydrogenation to get the final aromatic product. In the style

of the classic ‘borrowing hydrogen’ mechanism. Or more plausible perhaps is the hydrogenation of the C=N⁺ already present at this stage by the Mn-N-H₂ intermediate. Followed by another dehydrogenation of the remaining CH-OH...but then we still have the catalyst left as Mn-N-H₂, which takes us back to point 1 above.

Our Response: Thank you for your comments and suggestions. We numbered all the intermediates in the reaction mechanism. In the control experiment shown in **Fig. 5C**, 3-hydroxy-2-oxybutane **7** and **1a** reacted to afford **3a** in the absence of the Mn complex and base, with a yield of >99%. Therefore, we believe that the manganese catalyst only participated in the first dehydrogenation reaction of 2,3-butanediol **2a** to obtain 3-hydroxy-2-oxybutane **7** and did not participate in the subsequent reaction of 3-hydroxy-2-oxybutane **7** with *N,N'*-dicyclohexylurea **1a** to form **3a**. Based on this, we believe the mechanism follows the classic ‘borrowing hydrogen’ style, and the process where the hydrogenation of the C=N⁺ by the Mn-N-H₂ intermediate is followed by another dehydrogenation of the remaining CH-OH is unlikely. Therefore, we are more inclined to propose that after obtaining the dehydrogenation product 3-hydroxy-2-oxybutane **7**, the condensation between the ketoalcohol and the urea occurs first to form an iminium cation. The equilibrium of this species with an enamine–enol compound and a carbonyl compound explains the second condensation, which affords a five-membered cyclic compound. Finally, the cyclic iminium is reorganized into an enamine to afford the final imidazolone.

Some minor comments on phrasing:

P3 Line 26. Saying “numerous natural products” is a bit vague. Can you give a couple of specific examples.

Our response: Thank you for your comments and suggestions. We have modified the corresponding expression “imidazolones and their derivatives are valuable intermediates for synthesizing numerous natural products” to “imidazolones and their derivatives hold significant value in the synthesis of natural products, such as dibromophakelstatin and axinohydantoin” in the revised manuscript with track changes.

P3 line 40 “However, traditional synthetic methods cannot satisfy the increasing demand for imidazolones.”. I don’t understand this statement. Why cant all these methods provided ‘satisfy the increasing demand’ ?

Our response: Thank you for your comments and suggestions. We have modified the corresponding expression “However, traditional synthetic methods cannot satisfy the increasing demand for imidazolones” to “However, these methods often suffer from low atom efficiencies, poor functional group tolerance, and inferior regioselectivities” in the revised manuscript with track changes.

Line 43 “Alcohol is highly abundant...” it sounds like you’re talking about ethanol (or vodka!).

Rephrase

Our response: Thank you for your comments and suggestions. We have modified the corresponding expression “Alcohol is highly abundant, economical, ecofriendly starting material that can be produced from renewable bio-based feedstock” to “Alcohol molecules are highly abundant, economical, eco-friendly starting materials that can be produced from renewable bio-based feedstock” in the revised manuscript with track changes.

Line 60: “The previously reported catalytic systems require advanced specialized materials ...” sounds too vague. Be clear what the ‘advanced specialized materials’ are that you are concerned about.

Response: Thank you for your comments and suggestions. We have modified the corresponding expression “the previously reported catalytic systems require advanced specialized materials and are difficult to improve” to “the previously reported catalytic systems have problems such as difficulty in regulating the structure of active sites and extremely complex characterization” in the revised manuscript with track changes.

We thank all the reviewers for their comments and suggestions.

Thank you for your time and consideration. We look forward to hearing from you soon.

Best regards

Zhihui Shao

河南农业大学

中国 郑州 450002

Henan Agricultural University

Zhengzhou 450002, China

Dr. Zhihui Shao, *Associate Professor*

College of Tobacco Science, Henan Agricultural University

Zhengzhou 450002, P. R. China

shaozh21@henau.edu.cn

September 3th, 2025

Dear Editor:

Thank you very much for handling our manuscript titled “Mn-Catalysed Acceptorless Dehydrogenative Condensation of Ureas with 1,2-Diols for Synthesizing Imidazolones” (Manuscript ID: COMMSCHEM-25-0358A). We do appreciate the comments and suggestions from all the reviewers. Accordingly, we carefully revised this manuscript to address most of the comments and questions from the reviewers. Please find our statements and answers to all the suggestions and questions point by point as follows:

COMMENTS TO AUTHOR:

Reviewer: #1

Comments:

The authors successfully responded to all the points and in my opinion the manuscript should be published. I would encourage the authors to include some of the new experiences performed for responding to the referees (those not included, such as hydrogen detection) in the paper, at least in the supporting information.

Our response: We appreciate your recommendation of acceptance and helpful comments in the reviewing process and are pleased to have our manuscript be reviewed by you. According to

your suggestion, we have include some of the new experiences performed for responding to the referees (such as kinetic studies and hydrogen detection) in the supporting information.

Reviewer: #2

Comments:

The authors have conducted all the necessary revisions satisfactorily. I am happy for the paper to be now accepted to the Communication Chemistry.

Our response: Thank you very much for your valuable comments in the reviewing process, which have helped us to improve the quality of the whole manuscript. We sincerely appreciate you for recommending our manuscript be accepted by *Communications Chemistry*.

Reviewer: #3

Comments:

Overall, it appears all the corrections required have been completed. Some issues to consider below before publication:

Concerns:

Figure 4. Yield is down to 8% when N-Me catalyst used. But why does 8% product still form when the NH is apparently crucial for reaction to proceed. Does this suggest there's another mechanism at work here?

Our response: Thank you for your comments and suggestions. In this work, the *N*-methyl-substituted manganese catalysts **[Mn]-V** was synthesized to study the reaction mechanism of the hydrogen transfer process. If it is observed that the manganese catalysts **[Mn]-III** and **[Mn]-V** have comparable catalytic activities for the dehydrogenation condensation of **1a** and **2a** to imidazolone products, this is consistent with an inner-sphere mechanism without the metal–ligand cooperativity. If no imidazolone products were detected in the dehydrogenative condensation of **1a** and **2a** catalyzed by **[Mn]-V**, the result suggested that an outer-sphere hydrogen transfer mechanism via metal-ligand cooperation based on the “N-H effect” was the

reaction pathway for this manganese catalyzed dehydrogenative condensation reaction. Through experiments, we found that the use of **[Mn]-V** catalyst resulted in only a small amount of imidazolone product **3a** being produced. The obtained results demonstrate that the outer-sphere dehydrogenation mechanism is the major reaction pathway, and the N–H groups in the **[Mn]-III** catalyst are indispensable for the reactions between urea and diols to yield imidazolones or their derivatives.

In the mechanism text referring to Fig 5,

- Need to bold the number 6 in line 8
- Typo on line 9. 3-hydroxy-3-me....3c should be 3b

Our response: Thank you for your comments and suggestions. We have made the corresponding revisions in the revised manuscript with track changes.

In experimental section in Supp Info, need to include mass, % yield of the imidazole products. At the moment its just NMR data.

Our Response: Thanks for the suggestion from this reviewer. We have supplemented mass and yield of the imidazolone products in the Supporting Information.

I've started to take a detailed look at the NMR spectra in the Supporting Info and have some concerns.

- Fig 13: Looks OK
- Fig 15 doesn't look pure. There are clearly multiplet peaks around 2ppm not assigned. Also we can see the ppm range used for integrals raises concern. For the peak at 1.4ppm (integral 6.20) the ppm range doesn't look to include the full peak (more of this later on)
- Fig 17: good
- Fig 19 good, but again little impurity at about 1.9ppm multiplet
- Fig 21: again large peak at 2.2ppm not accounted for. Also, compare the 6.12 integral at 1.7ppm and the 2.19 integral at 2.5ppm. By eye, the former is not 3X greater integral than latter.

And looking at the narrowness of integral range on 2.5ppm peak suggests not all peak is included in integral reported. By contrast the ppm range on 1.7ppm peak has been exaggerated to include neighbouring signal.

I've scanned down later ones and they generally look OK, although some impurity peaks on occasion.

- Fig 68 look concerning again. Multiplet at 1.1ppm (4.37 integral) is clearly not integrating full peak. Compare to broad singlets at 2.2ppm (4.01 integral) and 1.5ppm (4.03 integral). By eye, the two latter signals are clearly lower integral than the former multiplet at 1.1ppm, despite both being labelled as having integral around 4.

My overall conclusion is that the reaction does work, but the authors have had some poor purification and chosen to include impure material in yield calculation. With some manipulation of integral values to match the expected numbers. It's important not to cut these corners and we need to either see the raw data files of the spectra picked out here or the authors go through thoroughly, repurify where necessary and clearly integrate the full NMR peaks.

Our response: Thank you for the careful review and valuable suggestions from the reviewers. We have examined the NMR spectra in the Supporting Information and re-performed the product separation for the reactions corresponding to 15, 19, 21, 23 and 68. We obtained purer NMR spectra and recalculated the yields corresponding to these reactions.

We thank all the reviewers for their comments and suggestions.

Thank you for your time and consideration. We look forward to hearing from you soon.

Best regards
Zhihui Shao